# Advanced informatics understanding of clinician-patient communication: A mixed-method approach to oral health literacy talk in interpreter-mediated pediatric dentistry

**Hai Ming Wong**[1]◉*, **Susan Margaret Bridges**[2]◉, **Kuen Wai Ma**[1]◉, **Cynthia Kar Yung Yiu**[1]‡, **Colman Patrick McGrath**[3]‡, **Olga A. Zayts**[4]‡

**1** Paediatric Dentistry, Faculty of Dentistry, The University of Hong Kong, Hong Kong SAR, China, **2** Faculty of Education/Centre for the Enhancement of Teaching and Learning, The University of Hong Kong, Hong Kong SAR, China, **3** Public Health, Faculty of Dentistry, The University of Hong Kong, Hong Kong SAR, China, **4** School of English, Faculty of Arts, The University of Hong Kong, Hong Kong SAR, China

◉ These authors contributed equally to this work.
‡ These authors also contributed equally to this work.
* wonghmg@hku.hk

**Data Availability Statement:** All relevant data are within the manuscript and its Supporting Information files.

## Abstract

### Background

In the clinical dental consultation, multi-party configurations predominate with the presence of parents/ primary caregivers in pediatric dentistry adding another layer of complexity. In managing child oral healthcare, parents/ primary caregivers are critical, especially in dental caries prevention. This study aimed (1) to identify the structure of oral health literacy (OHL) talk in interpreter-mediated pediatric dentistry and (2) to analyze interpreter contributions to the communication strategies: patient-centered direct interpreting (PC-DI), patient-centered mediated interpreting (PC-MI), clinician-centered direct interpreting (CC-DI), and clinician-centered mediated interpreting (CC-MI).

### Methods

Visual text analysis (VTA) of video recorded pediatric clinical consultations in Hong Kong utilized Discursis™ software to illustrate temporal and topical structures and their distribution across turns-at-talk. Conversation analysis (CA) was applied to analyze turn-taking of the identified OHL talk qualitatively. The mixed-method approach of combining VTA and CA was applied to analyze the patterns and features of the recorded OHL talk.

### Results

The conceptual recurrences of the 77 transcribed video recordings were plotted visually. CC- and PC-OHL talk were identified by the recurrence patterns of monochromatic and multi-colored triangular clusters formed by off-diagonal boxes, respectively. CA of interpreter-mediated turns supported earlier findings regarding patterns of MI in multilingual

**Funding:** This work received funding from the Research Grants Council General Research Fund (GRF) of the Hong Kong Special Administrative Region (Ref: 760112); URL: https://www.ugc.edu.hk/eng/ugc/index.html; funding recipient: SMB; the funders had no role in study design, data collection and analysis, decision to publish, or preparation of the manuscript.

**Competing interests:** The authors have declared that no competing interests exist.

adult dental consultations; however, the role of the interpreter in parent/ primary caregiver education and patient management was more distinctive in the pediatric dentistry.

## Conclusions

The mixed-method approach assisted in unpacking the complexities of the multi-party interactions, supported identification of effective communication strategies, and illustrated the roles of the dental professionals in initiating CC- and PC-OHL talk in pediatric dentistry. The intervention showed the implication of the professional education of evidence-based practices for clinicians in balancing agenda management and the communicative dimension of OHL with the help of VTA and CA in multilingual consultations.

## Introduction

### Multi-party in pediatric dentistry

Definitions of both health literacy (HL) and oral health literacy (OHL) began with a functional, patient-focused and diagnostic focus on general capacities "*to obtain, process, and understand basic health information and services needed to make appropriate health decisions*" [1]. Empirically, HL has been identified as a key factor linked with healthcare quality in terms of clinical effectiveness, patients' experience and safety across general medicine [2, 3], pediatrics [4] as well as oral health [5]. The appropriate patient HL enhances not only the patient's application of knowledge to their health decisions, but also their engagement, empowerment and ultimate motivation to follow a treatment plan [6]. For health professionals, patient HL assessments can assist in modifying or expanding health education content [7] and delivery and improving interactional quality. In pediatric dentistry, given the naturally limited OHL of children, parents/ primary caregivers' involvement in information seeking and provision becomes particularly significant to the sustainable transfer of oral healthcare messages from clinicians to children [8]. According to research into the functional OHL of primary caregivers, parental OHL is not only associated with child's oral health status [5], but is also as important as clinical treatment in maintaining good child oral hygiene [9]. Research into the communicative dimension of pediatric OHL has identified communication practices among dentists, dental surgery assistants (DSAs), child patients and their parents/ primary caregivers that can promote the quality of dental care [10, 11]. An informatics understanding of clinician-patient communication with a more communicative, interactive and socially engaged focus would be beneficial to promote OHL of patients. The OHL talk in the study was defined to be the health care communication to obtain, process, and understand basic oral health information of patients [12]. This multi-party configuration becomes even more complex and nuanced in a multilingual dental context such as Hong Kong where the DSA acts as interpreter and intercultural mediator [13] in both dentist-designated and semi-autonomous patterns of interpretation [14].

### Multilingual interpreter

The introduction of the notion of 'mediated' interpretation in healthcare, where the agency of the nurse interpreter is seen as central to effective multilingual consultations in general medicine [15] was in response to more linguistic, translation-based approaches to studies of direct interpreting (DI). Mason [16] made a distinction between notions of texts as discrete and

stable and the negotiated processes of unfolding interactions. Baker [17] identified the tensions between cognitive, linguistic code-based approaches and social/ interactive approaches to interpretation, noting the fluid, dynamic and negotiable aspects of context. Similarly, Setton [18] argued for "new theories of communication which shift the focus from linguistic stimuli to stimuli-in-contexts". Indeed, prior work in the context of multilingual adult consultations in dentistry in Hong Kong has highlighted the indeterminate nuances of mediated interpreting (MI) by DSA during consultations [14]. Hong Kong is an "Asia's world city" with a linguistic variation [19] due to its history and educational practices. The Hong Kong Basic Law and the Official Languages Ordinance declaimed that the English and Chinese are the official languages of Hong Kong of equal status. The trilingual code-switching in conversation between Chinese (Cantonese), Chinese (Putonghua), and English is common [20]. Multilingual interpreters are an essential resource in between English-speaking clinicians and the Cantonese-speaking patients for the multilingualism in the dentistry in Asia [13]. In terms of nurse autonomy and agency, Conversation Analysis (CA) of the triadic, adult corpus found the dentist engaged in designated and initiated patterns to dictate nurse-interpreted expansions under a shared agenda of patient care [14]. Research in ad hoc interpreting has raised concerns regarding the communication of the risks in medical complications [21] and the improvements in clinical shared decision-making [22].

## Communication strategies

'Clinician-centered' (CC) and 'patient-centered' (PC) [23] styles are widely used generic classifications in healthcare communication. While the latter has been viewed as more favorable with a perceived breaking down of barriers between clinicians and patients and has been linked to increased patient satisfaction [24], its instantiation in mediated interpreted healthcare has not been closely explored. The difference between the two strategies is most closely linked to participation structures [25], shared decision-making [26], egalitarianism [27] and relationship building [11]. Affectively, PC clinicians provide emotional support by showing empathy, attentiveness and understanding which promote patients' engagement, trust, understanding, satisfaction and motivation for self-care [28, 29]. Techniques and strategies such as teach-back and the interactive communication loop [30] and motivational interviewing [31] are some methods applied to enhance PC approaches to clinical communication. Typically, these sequences of talk were predominantly related to interactive counseling for parent/ primary caregiver oral health education and included both DI and MI [14, 15].

## Novel mixed-method approach

However, there are challenges in the medical information extraction from the clinician-patient communication (dentists and DSAs vs child patients and their parents/ primary caregivers in pediatric dental practices). The computer-aided technology development [32] is essential for the analysis of mutimedia such as voices [33] and videos [28]. A visual text analysis (VTA) tool application such as Discursis™ [34, 35] (https://www.itee.uq.edu.au/research/projects/discursis) to transcripts of medical consultations is a novel approach to graphically illustrate structural patterns of clinician-patient communication from a sequential and interactional perspective. Research using CA has contributed to understanding the social organization of clinician-patient interactions through uncovering patterns of recorded and transcribed talk to establish both the construction and understanding of action [36]. A mutual support of the VTA and CA can be achieved by the mixed-method approach. One of the challenges in the information extraction is the time required for the identification of OHL-related talk prior to CA analysis. At the same time, the interpretation of participation patterns is required for the

information extraction from the conversations. The VTA visual representation can support the application of CA by identifying the turn structures more effeciently for ensuing CA micro-analysis of the VTA results. Hence, the aims of the study were to identify '*where*' the OHL talk occurred through visualizing sequences and participation patterns, and to explore '*how*' multi-party, multilingual pediatric dental consultations were interactionally managed by participants through a focus on participants' understandings of one another's actions as evident in taking up (or not) what is heard [37] by adopting the mixed-method approach of combining VTA and CA. The study also aimed at providing an evidentiary account of nuanced understandings of the communicative/ interactional domain of OHL to support more effective multilingual oral health care exchanges between the clinical team, the child patients and their primary caregivers [38], specifically in interpreter-mediated pediatric contexts.

## Materials and methods

### Data collection

The video recordings of routine patient visits at the pediatric clinic in a dental teaching hospital in Hong Kong examined here is a sub-set of a larger video-recorded corpus (n = 199) investigating clinician-patient communication in an era of global movement (GRF 760112). Ethical approval was obtained from the Institutional Review Board of the University of Hong Kong/ Hospital Authority Hong Kong West Cluster (IRB: UW 12–068). The dental team and pediatric patients/ primary caregiver (parent) dyads provided written informed consent to video recording of clinical consultations. The information of language-in-use, gender (patients and parent/ primary caregivers only), and age (patients only) was provided by dental team and/or parent/ primary caregivers via a self-complete questionnaire before video recording. In order to ensure that the analyzed videos included all the desired elements such as "pediatric consultations", "multi-party", and "multilingual interpreter", three criteria were set for the video selection from the corpus: (1) The age of patient was under 18 years old (pediatric dental consolations); (2) patient was accompanied by a parent/ caregiver (multi-party); and (3) there were Cantonese speaking children and their caregiver/ parent, and a clinical team dyad of non-local clinician (English language-in-use) with the same DSA as the interpreter (multilingual interpreter). Only the recorded videos that fulfilled all the criteria were transcribed and further analyzed in the study.

The recorded multilingual (Cantonese-English) conversations were transcribed using Transana™ software version 2.53 with Jeffersonian notation (see S1 File). All Chinese characters were parsed into English. Transcriptions and translations were verified by the team members. Data analysis of the transcribed consultations consisted of two phases.

### Phase I: Visual text analysis (VTA)

The VTA using Discursis™ software provided graphic overviews of participation patterns and areas of topical coherence, or the lack thereof, through examination of turn-taking and distribution of content over the time course of each clinical consultation [34, 35, 39, 40]. Discursis™ input transcripts consisted of English-only translations and excluded transcriber notes and CA transcription conventions.

**Discursis™ plots.** Initial, 'long time scale plots' as overall conceptual recurrence plots were generated to provide a global overview of both the structure and content of each consultation. Fig 1 illustrates the long-time scale recurrence pattern of a Discursis™ output for Consultation 1. Common topics such as "tooth" and "mouth" that are introduced in the initial phase are repeated and linked across the entire consultation. Time scale, direction and interaction type were the three principle dimensions used to define the multi-participant metrics of the

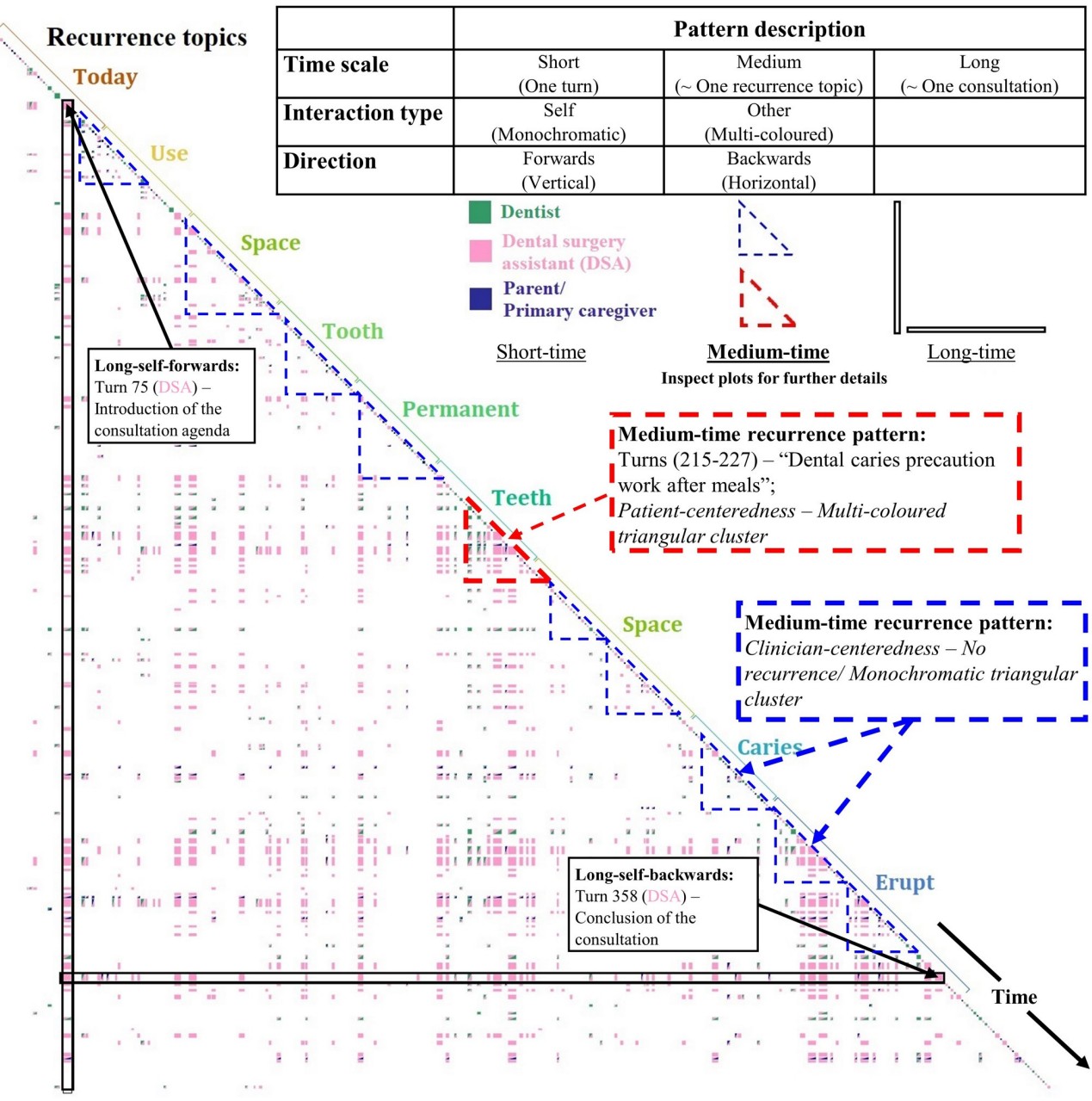

**Fig 1. Overall conceptual recurrence plot of a pediatric dental consultation.**

recurrence patterns [35]. These indicated whether the recurrences were self/same speaker or other-recurrence, either following a turn (forward-direction) or recurrences related to previous turns (backward-direction) within short, medium or long period of time. Participation patterns are visualized from the color distributions and the patterns of conceptual recurrence plotting with the color, and the size of boxes in the plots corresponding to the speaker and volume per turn. The color of the lower right corner of the 'off-diagonal' boxes represents the speaker of the turn, while the color of the upper left corner of the off-diagonal boxes represents the speaker who shares the repeated topics or concepts. Therefore, a single or a two-colored

off-diagonal box refers to the situation where a speaker repeats his or her own topics, or a speaker repeats the topics or concepts from another speaker respectively. Discursis™ inspect plots (for example, see Figs 2–7) provide the zoom-in views of the medium-time recurrence patterns from the overall plots. The qualitative analysis of the turns-at-talk among the multi-party participants was conducted in Phase II using CA.

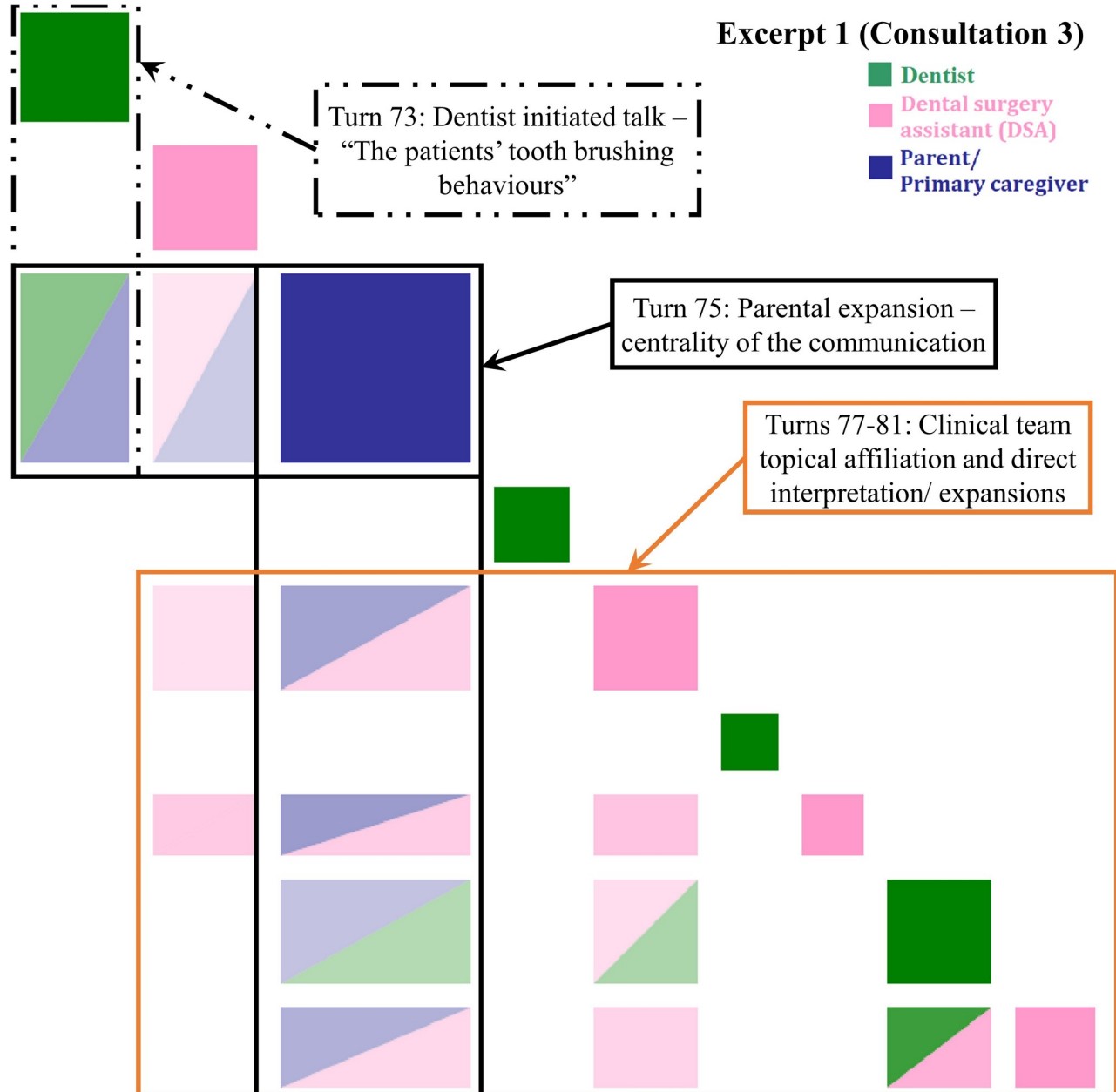

**Fig 2. Inspect plot of turns 73–81 in Consultation 3 indicating patient-centered direct interpreting.** "The patient's tooth brushing behaviors"—an eight-year-old girl is here for follow-up and a possible accident causing mal-alignment of teeth.

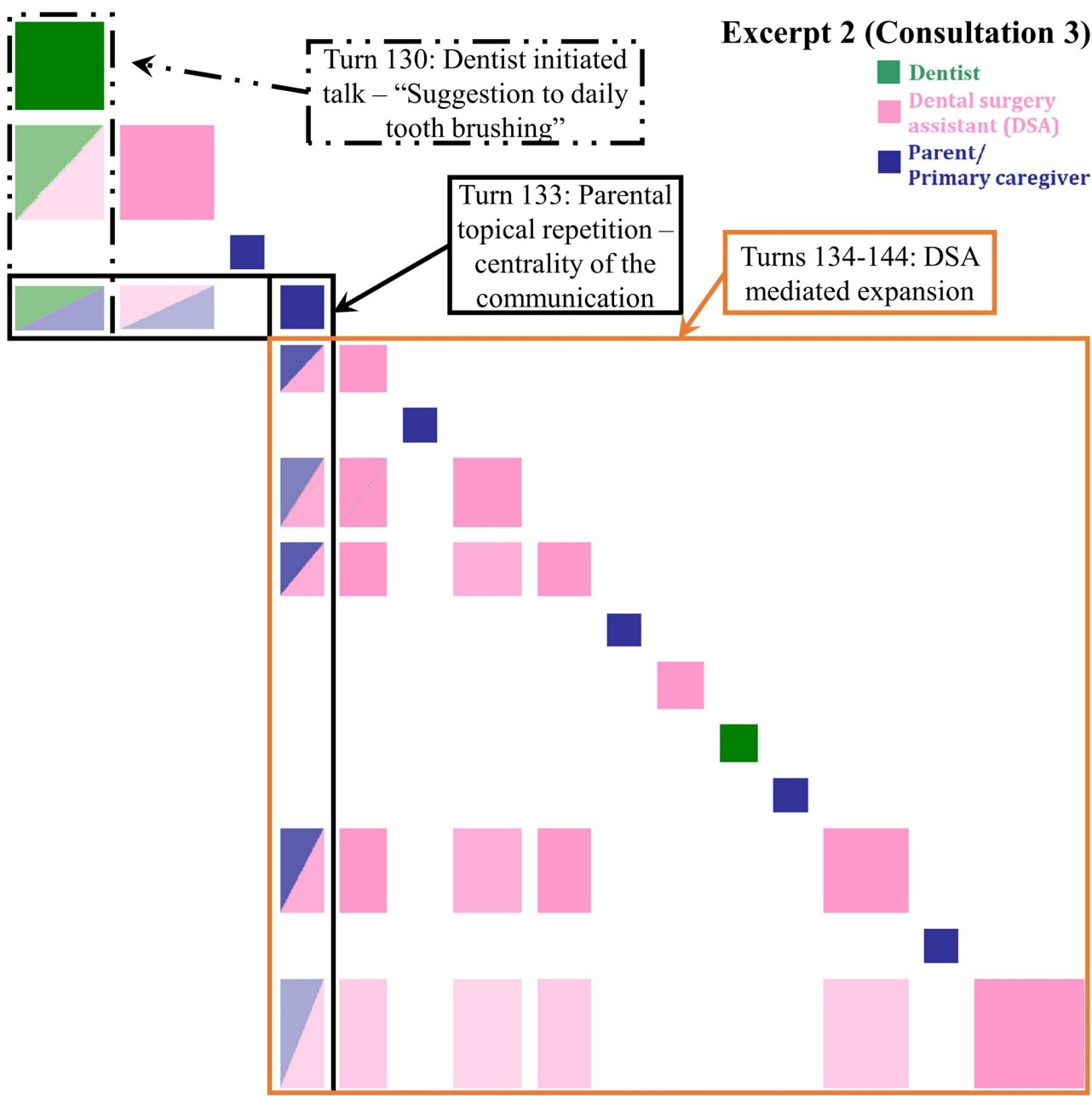

**Fig 3. Inspect plot of turns 130–145 in Consultation 3 indicating patient-centered mediated interpreting.** "Suggestion for daily tooth brushing"— an eight-year-old girl is here for follow-up and a possible accident causing mal-alignment of teeth.

## Phase II: Conversation analysis (CA)

Once sequences of interest were identified visually, the original, multilingual transcriptions with Jeffersonian notation were used for micro-analysis using CA. "Turn taking" [41] is a systematic organization in CA to express the participants' utterance (a speak of participants at a time in alternating turns) via the construction and allocation of turns from the field recorded conversation. In practice, the discovery of the turns-at-talk allowed the researchers to analyze the communication processes for constructing contributions, responding to others, and

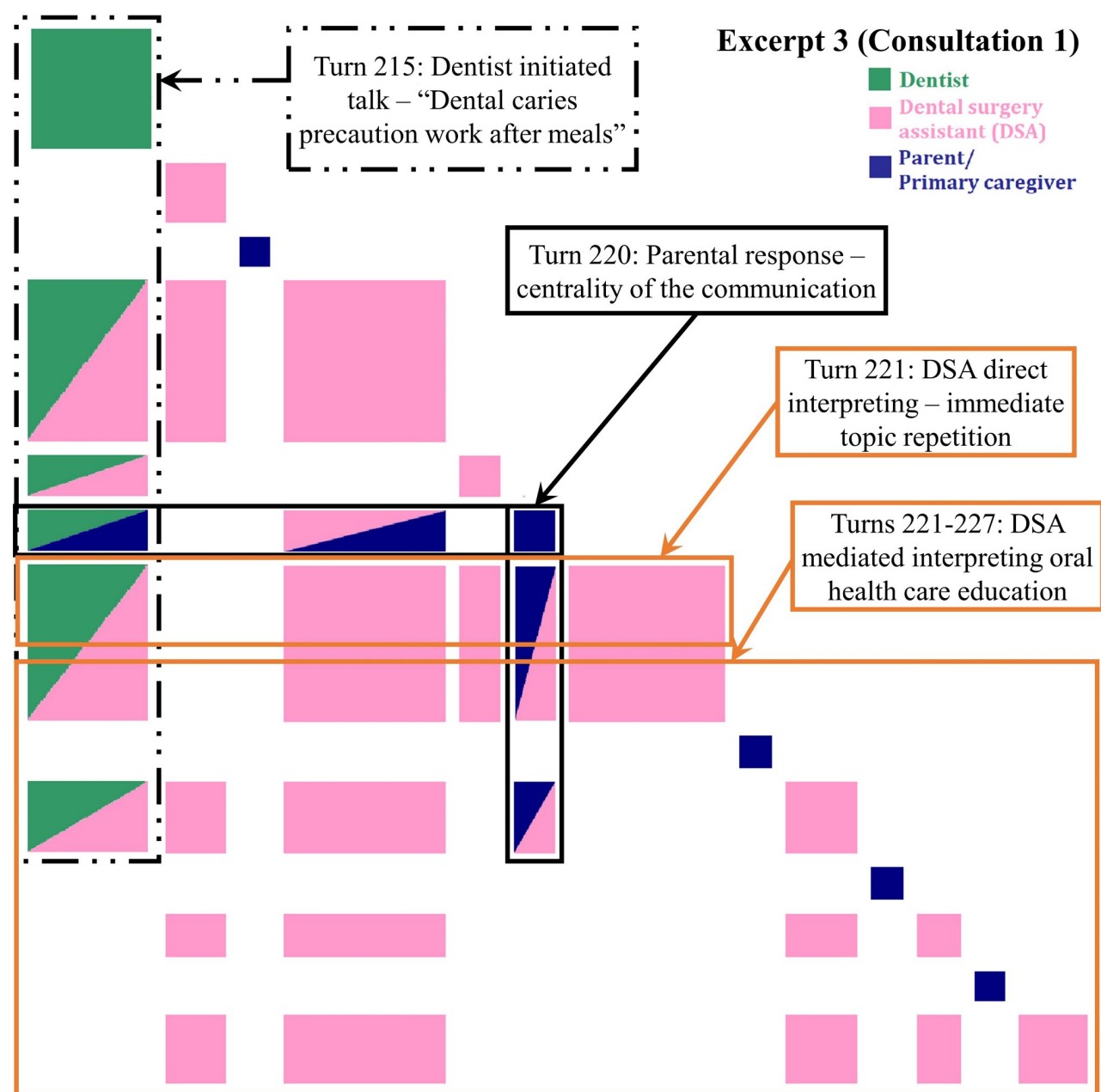

**Fig 4. Inspect plot of turns 215–227 in Consultation 1 indicating patient-centered direct and mediated interpreting.** "Dental caries prevention work after meals"- an eight-year-old boy previously treated under general anesthesia has erupting permanent teeth.

changing of speakers rather than consider the conversation as a whole [42]. As a qualitative approach, the aim of CA is to uncover patterns of naturally occurring recorded talk to establish both the construction and understanding of action [36]. In analyzing medical interactions, CA supports identification and micro-analysis of both the contingent and collaborative nature of face-to-face medical interactions [37]. As in the team's prior study of MI in adult dental consultations, we explored recipient design [43–45] to understand how interpretations emerged or were constructed during pediatric dental consultations.

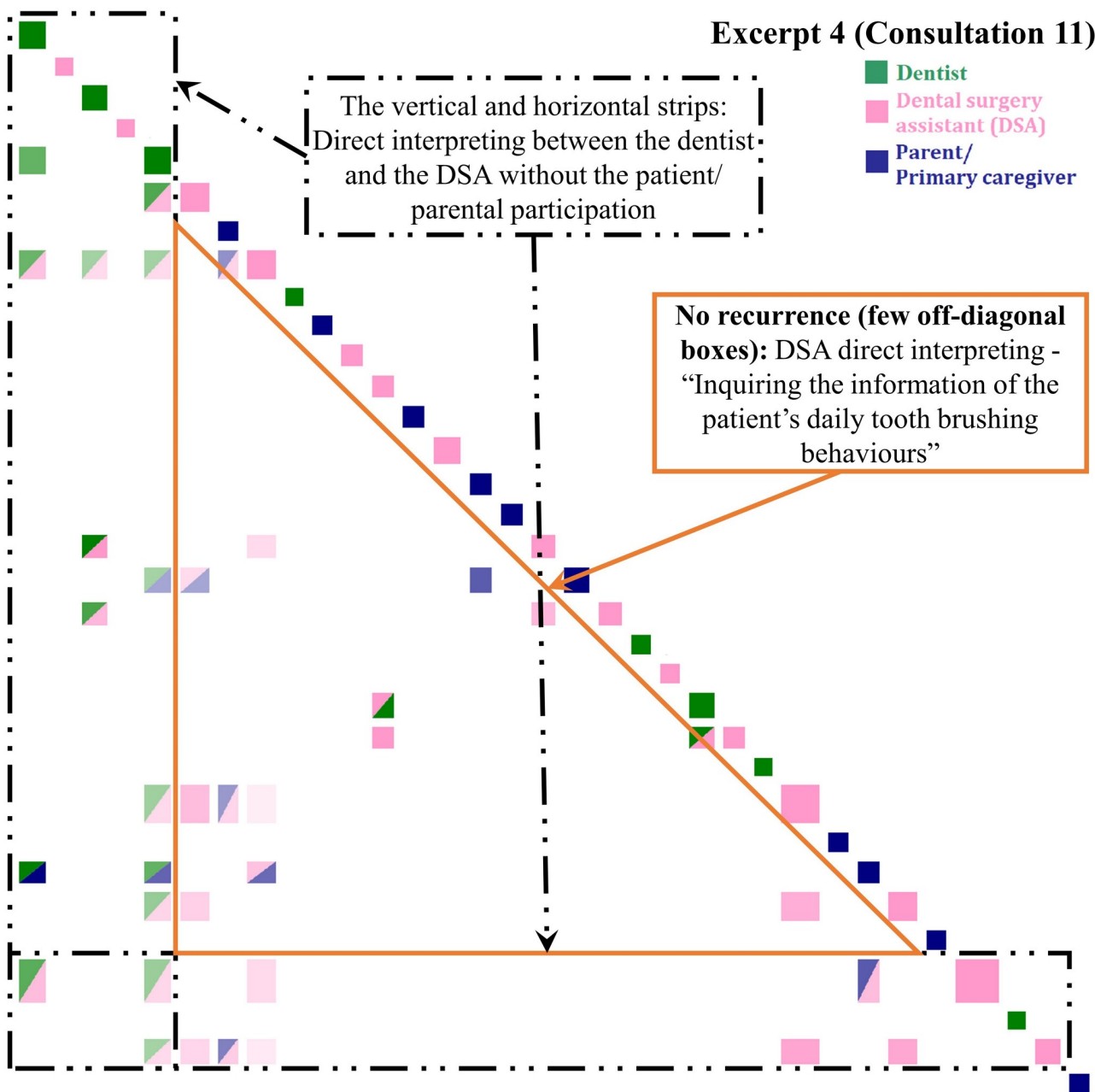

**Fig 5. Inspect plot of turns 34–66 in Consultation 11 indicating clinician-centered direct interpreting.** "Inquiring about the patient's daily tooth brushing behaviors"—a seven-year-old boy is here for his six-month review after previous treatment under general anesthesia.

## Results

### Statistical description of data

The average length and turns-at-talk of the 77 transcribed video recordings were 12.62 minutes and 290 turns respectively. Patients comprised 41 girls and 36 boys (mean 7.39 years, standard deviation = 3.40 years). Parent/ primary caregivers comprised 54 females and 23 males with Cantonese as their language-in-use. The dental team comprised the specialist pediatric dentists (English language-in-use) and DSAs (Cantonese/ English languages-in-use) with periodic inclusion of a consulting senior dentist (English/ Cantonese language-in-use). All the

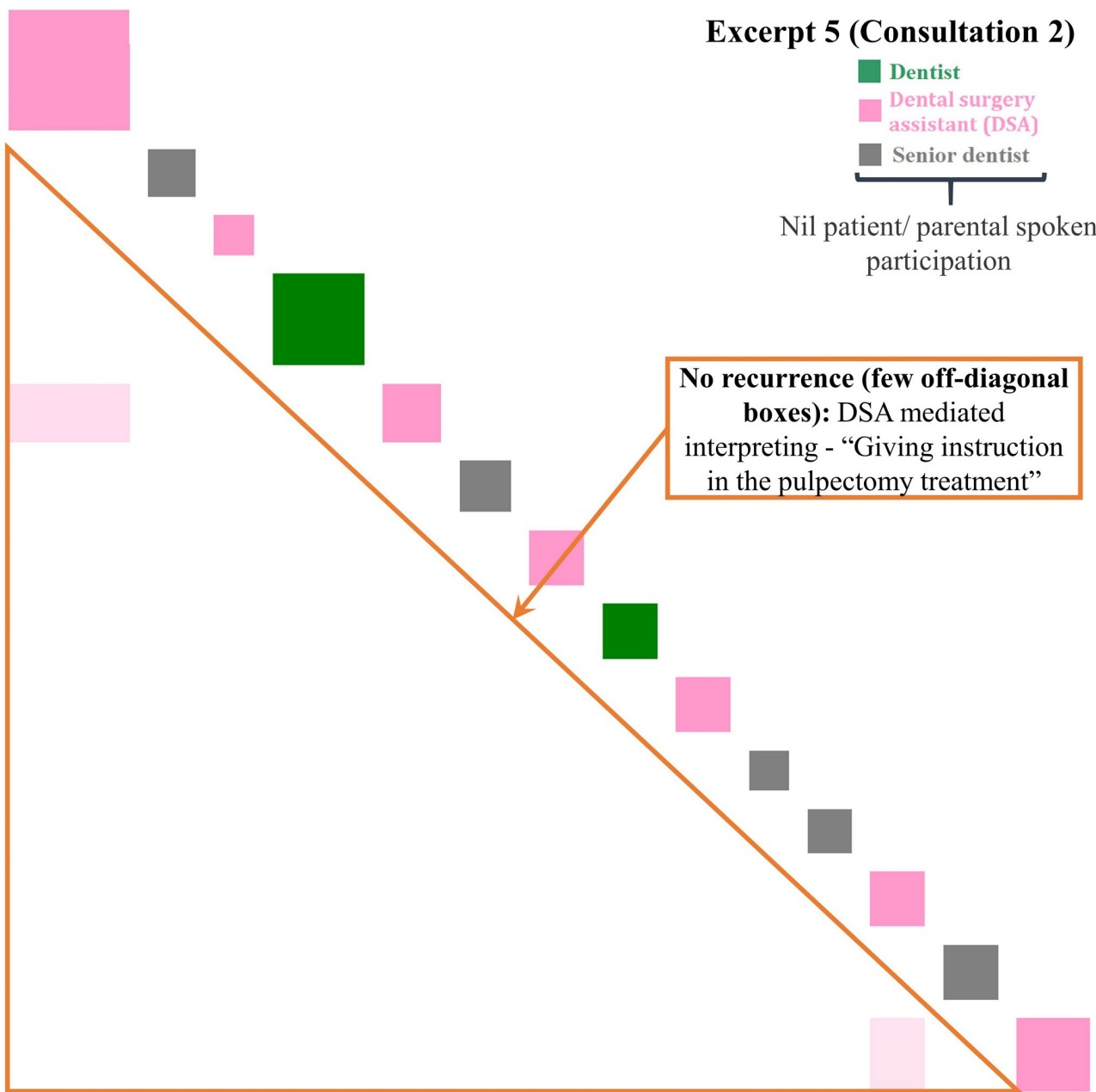

**Fig 6. Inspect plots of turns 18–31 in Consultation 2 indicating clinician-centered mediated interpreting.** "Giving instructions for the pulpectomy treatment"—a seven-year-old boy has missing permanent teeth.

transcripts of the excerpts in the inspect plots (Figs 2–7) were shown in S2 File. It is recommended that the plots were read together with the relevant text transcripts in the supporting information to have a better understanding of the results.

### Patient-centered (PC) communication

PC communication was identified where the recurrence pattern displayed multi-colored (i.e. multi-party interactive discussions) triangular cluster indicating overlapping topical and temporal interactions.

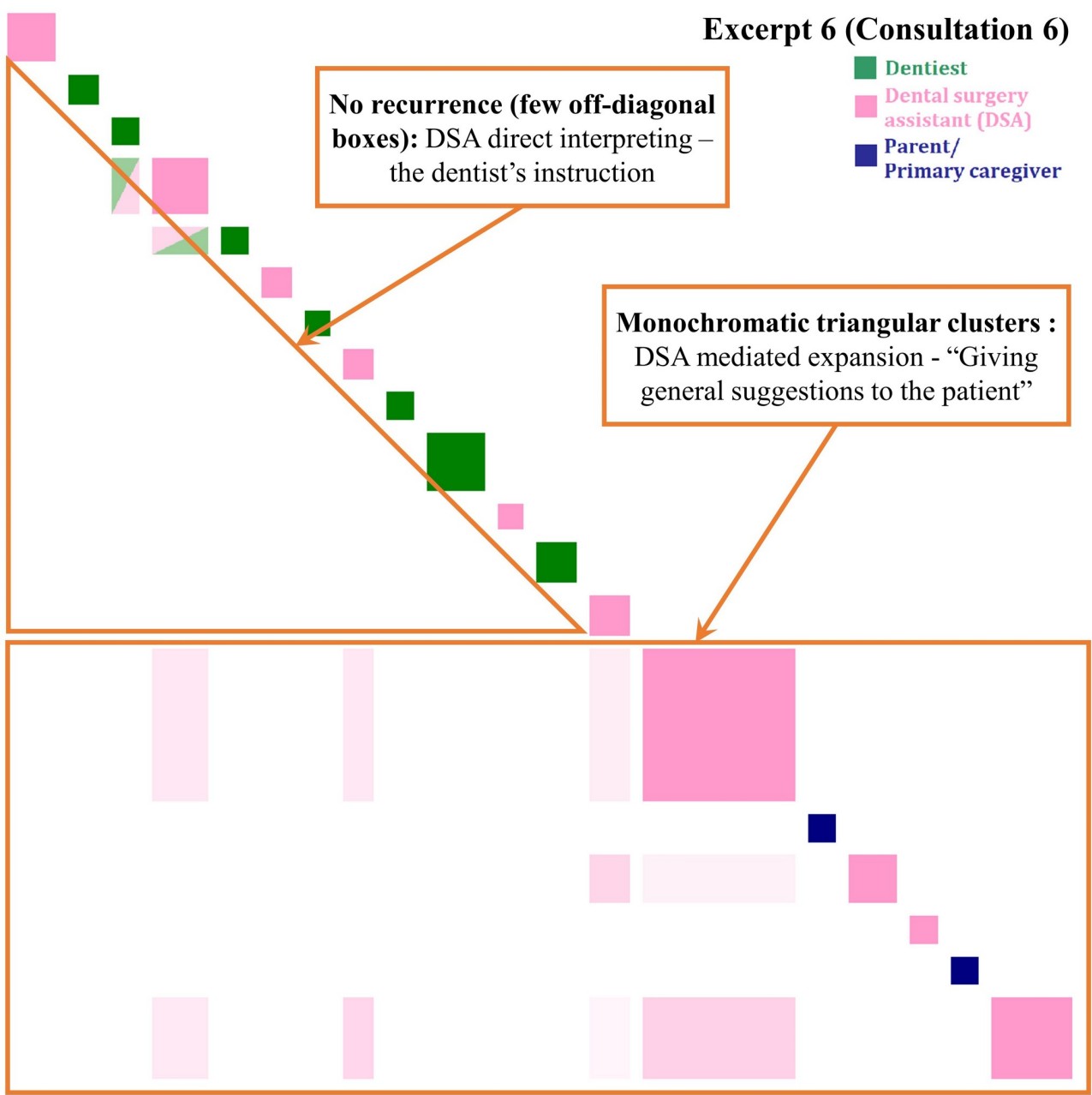

**Fig 7. Inspect plot of turns 127–146 in Consultation 6 indicating clinician-centered direct and mediated interpreting.** "Giving general suggestions to the patient"–a seven-year-old-girl is brought in as she has caries.

**Patient-centered direct interpreting (PC-DI).** Fig 2 highlights the significance of the parent's turn introduced the new topic of electric tooth brush and child autonomy. The visualization illustrates the centrality of this parent turn as topically relevant in the way that it responds to prior turns (green-blue and pink-blue off-diagonal boxes) and was taken up by the dentist and DSA in subsequent turns (blue-pink and blue-green off-diagonal boxes). The topic of tooth brushing behavior was initiated by the dentist's question after noticing the patient's poor oral hygiene status. The DSA translated the question and directly interpreted the parental response. Finally, the dentist affirmed the DSA's interpretation. The recurrence pattern of a

multi-colored triangular cluster occurred when participants reached agreement regarding the parent's assistance in encouraging the child to use her electric toothbrush.

**Patient-centered mediated interpreting (PC-MI).** In the adult patient corpus, we had identified the patterns with evidence of direct, summary interpretations; dentist-designated interpreter-mediated expansions (Patterns 1 & 2); and unprompted, autonomous expansions (Pattern 3) by the interpreter in dentistry in Hong Kong [14]. In this pediatric corpus, the PC interaction in Fig 3 is illustrative of the Pattern 3. Noticing the eruption of the patient's permanent molar, the dentist suggested that the patient should brush inwards. The DSA heard the dentist's advice as transition relevant to designation to interpret and took up the expansion to the parent. The advice was also repeated by the parent. However, the dentist retained control of the interpreting through the checking, *Alright*? The DSA heard both the dentist checking and the parental minimal response as transition relevant to providing elaboration and expanded with further oral hygiene. Similarly, the blue box of the parental topical repetition linked with most of the turns in this recurrence pattern showing the centrality compared with Fig 2. Although the repetition did not provide any new ideas or information, it signaled the parent's acknowledgment of the clinician's ideas, represented in the green-blue and pink-blue off-diagonal boxes. The subsequent blue-pink and pink off-diagonal boxes illustrate the DSA's turns linked with the concept 'brush' in the parent's repetition. She emphasized further the reason and the method of how the parent might assist functional HL education during her final expansion with the introduction of the lexical items ('plaque', 'gum sides' and 'caries'). Additional use of the 'teach-back' strategy may have been a further opportunity for parent/ primary caregiver functional OHL development.

**Patient-centered direct interpreting and mediated interpreting (PC-DI and MI).** The interpreting undertaken in Fig 4 also follows a dentist designated pattern [14] with direct verbal request signaled at turn 215 with *See if. . ..* In this excerpt, lodged food was found in the gap between the patient's teeth during dental scaling by the dentists. The dentist suggested that more attention should be paid to the risk of caries caused by food debris, especially when permanent teeth erupt. The DSA then suggested that the parent remind the patient to brush his teeth or drink water to remove food debris (direct interpreting). This led to ensuing the DSA's and parent's turns focusing on the concepts of 'caries', 'food', 'school' and 'brush' in the dentist's advice which indicated by the green box in Fig 4. The subsequent green-pink and green-blue off-diagonal boxes illustrate the topical relevance of the dentist's advice.

In turn 220, the parent confirmed that the child would drink water after his meals. Although the parent's response was brief, *drink (.) did drink after eating*, it also created the centrality of the communication. The green-blue and pink-blue off-diagonal boxes were topically relevant to the word 'drink' in the dentist's advice and the DSA's suggestion. After that, the DSA immediately provided the alternative suggestions of using a tooth pick or flossing (DSA initiated mediated interpreting) [14], generating the blue-pink off-diagonal boxes. The DSA initiated the unprompted expansion aligned to her oral health education role by emphasizing that caries would still be caused if food debris cannot be removed by drinking water. The pink off-diagonal boxes form the self-type recurrence pattern. This illustrates that the DSA heard what the parent had said and signaled this immediately by responding with advice that was topically relevant and supportive of the clinical team's agenda.

## Clinician-centered (CC) communication

CC communication was identified where the recurrence pattern displayed few off-diagonal recurrence boxes and predominantly monochromatic, triangular clustering with large boxes indicating dominant turns-at-talk in an information-giving role.

**Clinician-centered direct interpreting (CC-DI).** This excerpt was selected as example because it might be easily misjudged as PC communication. Although there were the multi-colored off-diagonal boxes in Fig 5, the clinician-patient interactions were limited. In Excerpt 4, dentist designated a question to the parent about the patient's daily tooth brushing behaviors. After omitting the DI of the bilingual DSA between the English-dominant dentist and the Cantonese dominant patient's parent, the recurrence pattern of few recurrence boxes is identified for little topic relevance in the OHL talk and lack of multi-party interactions.

**Clinician-centered mediated interpreting (CC-MI).** In Fig 6, the DSA self-initiated the OHL education (MI) in Cantonese and explained the pulpectomy treatment step by step using lay language [46] to the child patient in silent. Only two off-diagonal boxes were created due to the recurrence of the common words ('tooth' and 'mouth') during the pulpectomy treatment. The recurrence pattern of few off-diagonal boxes indicates that the concepts ('cream', 'ring', 'rain coat', and 'suction', etc.) in the treatment procedures were discrete. The limited similarities of the concepts reduced the occurrences of off-diagonal boxes. In addition, no boxes corresponding to the patient or her parents, as well as the lack of multiple colors, reflects the patients' limited participation in this CC communication.

**Clinician-centered direct interpreting and mediated interpreting (CC-DI and MI).** Fig 7 illustrates the DSA's ability to switch between interpreting patterns. Structurally, two parts are evident here, with the transition relevance placed at the dentist departure. DI illustrate the coordinated activity of moving the child into correct placement on the chair and gaining access to the oral cavity for examination. The recurrence pattern of few off-diagonal boxes echoes instructions-giving patterns. Later, the dentist provided a summary of findings and his decision to discharge with his concern. The dentist designating a request for interpreting is followed by a transition to DSA mediated expansions.

Then, the DSA began an extended oral health education sequence with minimal response tokens (i.e. 'ah', 'ok', 'oh') by the parent. The long explanations of the general anesthesia treatment and dental hygiene maintenance enlarged the turn boxes in the recurrence pattern of a monochromatic triangular cluster representing the repetition of 'tooth'.

## Discussion

### Comparison with prior work

The micro-analytic work of this paper has sought to convey the qualities of patient-centered care in interpreter-mediated pediatric dentistry. The mixed-method approach to interactional analysis supported prior findings from interpreted adult dental consultations with evidence of direct, summary interpretations; dentist-designated interpreter-mediated expansions (Patterns 1 & 2); and unprompted, autonomous expansions (Pattern 3) by the DSA [14]. Evidence from this study was the different functions of clinician- and patient-centered talk in a multi-party, mediated interpreting context involving children and their parents/ caregivers. VTA highlighted the 'where' of participation patterns while CA of turn design identified the 'how'. Overall, clinicians were found to be dominant in instruction and explanation giving; however, patient participation was central to many of the interpreter-mediated turns-at-talk.

### Principal results

With regard to the *1st* research question of '*where*' the OHL talk occurred, the clinician-dominated and multi-party participation patterns were successfully visualized by the recurrence patterns of monochromatic and multi-colored triangular clusters in the Discursis™ plots, respectively. The former pattern was preliminarily identified to be CC communication style, while the latter was PC communication style. Conventionally, it was hard to quantitatively

assess the level of interactive engagement of participants in conversations. This increased the difficulties in identifying different communication styles. The study results showed that the application of the VTA can be an alternative to the preliminary identification of the communication styles, and a valuable approach of the PC communication studies. Subsequently, the in-depth CA of the turns-at-talk among the multiparty participants (dentists and DSAs vs child patients and their parents/ primary caregivers) gave the knowledge of '*how*' multi-party, multilingual pediatric dental consultations were interactively managed–the *2nd* research question. The two communication styles (PC and CC) can then be further divided in to three strategies (-DI, -MI, and -DI and MI). DI illustrated how the DSA (multilingual interpreter) cooperated with dentists, and MI illustrated how the interpreter tried to build a connection with the patients/ primary caregivers. The mix of DI and MI further showed how the multi-party, multilingual pediatric dental consultations were constructed by the interactions of the participants. The importance of multilingual interpreter was demonstrated in both unveiled communication strategies. Evident from this study was the different functions of CC and PC talk in a multi-party, mediated interpreting context involving children and their parents/ caregivers in pediatric dentistry (Fig 8). Completing dental treatments such as a pulpectomy and taking clinical dental radiographs while managing child patients are a race against time. Giving patients clear didactic instructions and explanations during ongoing treatment depend on the clinicians' medical skills and experience. The conversational dominance by the clinical team may be viewed in these situations as necessary to the smooth delivery of dental treatment and oral health education. In addition, interpreter mediated interactions also typically included functional literacy strategies in supporting parents/ primary caregivers with health system navigation [47]. These included avoiding the unnecessary use of medical terms and complex language [48], for example, replacing the medical term 'topical anesthetic gel' with the lay term 'cream'. From a communicative health literacy perspective [47], CC autonomy can be viewed as supporting rather than hindering effective communication, particularly when seen in light of other discourse-based research in the mediated interpreting field of healthcare which sees interpreters taking greater agency in providing appropriate healthcare instructions [15].

It has been argued that quality dental care is enhanced through PC care [11, 49]. From a health literacy perspective, conceptual and operational clarity of terms are gained as the result of responding to patients' questions, expressing opinions, stating preferences, and eliciting treatment options [50]. In general medicine, studies have indicated that communication with more information in a personally relevant context and with more expression of the patients' voice in the interactivity reduced demands on patient literacy and facilitated effective health care exchanges [48]. In the results above, the centrality of the OHL talk was linked to the patients' turns [51]. The multi-colored triangular clusters showed that patient participation in providing, seeking and verifying information on topics directly connected to what clinicians had said enhanced the extent to which clinician listened carefully to and understood patients with the aim of enhanced decision-making [52, 53]. In the complex multi-party and multi-lingual context in pediatric dentistry, a more egalitarian participation structure was helping the clinical team to align their decisions more closely with their patients' and parent/caregivers' needs, concerns, interests and preferences.

## Limitations

While all 77 video recordings were transcribed and analyzed using Discursis™, quantitative reports of all the possible communication structures associated with either clinician-centered or patient-centered communication approaches in interpreter-mediated clinical communication were not possible given the qualitative reports focus on how these interactions

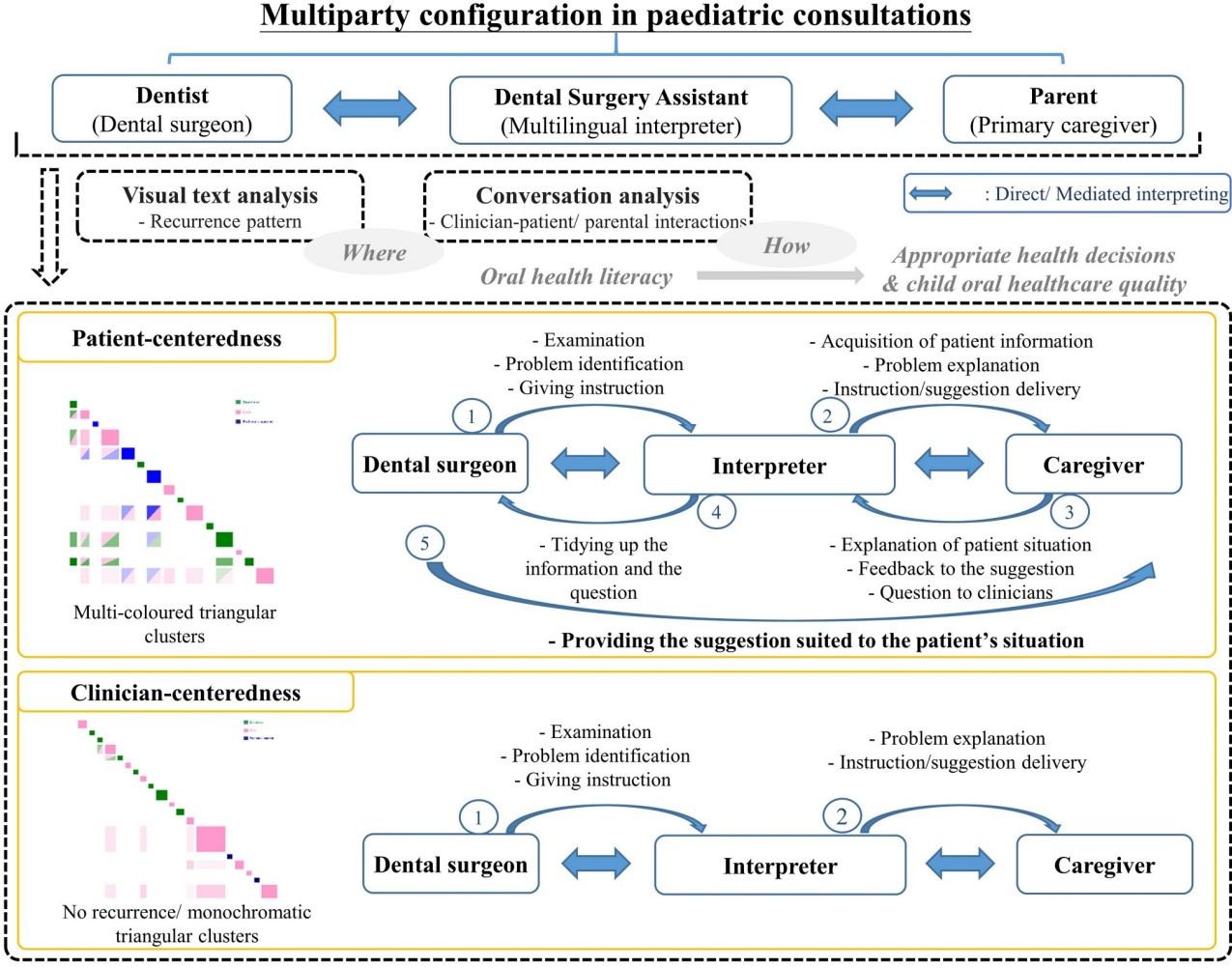

**Fig 8. Summary of the mixed-method approach to oral health literacy talk in interpreter-mediated pediatric dentistry.**

were formulated through the turns-at-talk. Therefore, only the recurrence patterns of selected occurrences were discussed. Non-verbal aspects of interactions were inevitably excluded from this study resulting in missing of some conversational structures in the conceptual recurrence plots. For example, patients answered the questions with the non-verbal action of nodding.

The main focuses of the study were to investigate the recurrence patterns of OHL talk in interpreter-mediated pediatric dentistry and to qualitatively explore the content of the talk. "Language-in-use" of the multi-parties was the major criterion in the selection of the OHL talk including a multilingual interpreter. Despite the randomness of patient background, the same DAS (multilingual interpreter) was involved in all the selected videos to reduce the effects of clinicians' background. However, the effects of parents and patients' socio-economic status [54] such as educational level on communication strategies, or the effects of communication strategies on the outcomes [3, 4] such as their satisfaction, were not investigated and discussed in this study. Those effects need to be further quantitatively investigated for designing a guideline on good health communication practices.

### Next steps

In terms of clinical education, the application of the VTA and CA holds the potential of future communication training for dental professionals. VTA would help reduce the time required for reviewing lengthy OHL videos and transcripts by supporting identification and extraction of key turns for in-depth analysis. In an ideal situation, a better clinician-patient communication can be achieved by all multi-parties with higher educational levels. Although the educational levels of patients or their parents are not controllable, the communication skills of dental professionals are educable. The popularization of the use of VTA is hence essential for dental professional's communication training in their communication skills on the intervention and oral health delivery works to obtain, process, and understand basic oral health information of patients. The future oral health professionals' education with evidence-based practices [55, 56] will be benefited from the VTA and CA application in defining effective communication strategies.

## Conclusions

With the help of VTA tools, the rendering of the content and structure of clinician-patient interactions in a digital record becomes analytically valuable. The mixed-method approach in interpreter-mediated pediatric dental consultations has supported not only our prior proposition regarding the centrality of the clinician in interpreter-mediated consultations but also corroborated the patterns of mediation identified in adult dentistry [14]. Nonetheless, analysis of these pediatric consultations indicates a greater level of clinician agency and autonomy in patient management and parent/ primary caregiver oral health education than in the adult dentistry study. The advanced informatics understanding of clinician-patient communication gave the insights of the right balance between CC- and PC- strategies. This study of mediated interpreting in multilingual dental consultations highlights the need, in a globalized era, for deeper and more nuanced understandings of multilingual teamwork in pediatric dentistry. The feasibility of the VTA in the professional education of evidence-based practices was also disclosed.

## Supporting information

**S1 File. Transcription conventions with Jeffersonian notation.**
(DOCX)

**S2 File. Transcripts and patient information.**
(DOCX)

## Acknowledgments

The research team would like to thank the clinicians, dental surgery assistants and patients who have participated in this study, and Dr Alexandra BH Chong for research assistance.

## Author Contributions

**Conceptualization:** Hai Ming Wong, Susan Margaret Bridges, Kuen Wai Ma, Olga A. Zayts.

**Data curation:** Hai Ming Wong, Susan Margaret Bridges, Kuen Wai Ma.

**Formal analysis:** Hai Ming Wong, Susan Margaret Bridges, Kuen Wai Ma.

**Funding acquisition:** Susan Margaret Bridges.

**Investigation:** Hai Ming Wong, Cynthia Kar Yung Yiu.

**Methodology:** Hai Ming Wong, Susan Margaret Bridges.

**Project administration:** Hai Ming Wong, Susan Margaret Bridges.

**Resources:** Cynthia Kar Yung Yiu.

**Software:** Kuen Wai Ma.

**Supervision:** Hai Ming Wong, Susan Margaret Bridges, Cynthia Kar Yung Yiu, Colman Patrick McGrath, Olga A. Zayts.

**Validation:** Colman Patrick McGrath.

**Visualization:** Kuen Wai Ma.

**Writing – original draft:** Hai Ming Wong, Susan Margaret Bridges, Kuen Wai Ma.

**Writing – review & editing:** Cynthia Kar Yung Yiu, Colman Patrick McGrath, Olga A. Zayts.

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
