## [Decision Letter · Decision Letter 0]

6 Jan 2020

PONE-D-19-27918

Advanced informatics understanding of clinician-patient communication: a mixed-method approach to oral health literacy talks in interpreter-mediated pediatric dentistry

PLOS ONE

Dear dr. Wong,  

Thank you for submitting your manuscript to PLOS ONE. After careful consideration, we feel that it has merit but does not fully meet PLOS ONE’s publication criteria as it currently stands. Therefore, we invite you to submit a revised version of the manuscript that addresses the points raised during the review process. In particular, the points raised by reviewers about the method and results-section should be improved upon before the manuscript can be considered for publication in PLOS ONE.

We would appreciate receiving your revised manuscript by February 10th. To enhance the reproducibility of your results, we recommend that if applicable you deposit your laboratory protocols in protocols.io, where a protocol can be assigned its own identifier (DOI) such that it can be cited independently in the future. For instructions see: http://journals.plos.org/plosone/s/submission-guidelines#loc-laboratory-protocols

We look forward to receiving your revised manuscript.

Kind regards,

Barbara Schouten

Academic Editor

PLOS ONE

2. Please provide further details in order to justify why only 77/199 of the larger dataset were included.

Reviewers' comments:

Reviewer's Responses to Questions

**Comments to the Author**

1. Is the manuscript technically sound, and do the data support the conclusions?

Reviewer #1: Partly

Reviewer #2: No

2. Has the statistical analysis been performed appropriately and rigorously? 

Reviewer #1: N/A

Reviewer #2: No

3. Have the authors made all data underlying the findings in their manuscript fully available?

Reviewer #1: No

Reviewer #2: No

4. Is the manuscript presented in an intelligible fashion and written in standard English?

Reviewer #1: Yes

Reviewer #2: No

5. Review Comments to the Author

Reviewer #1: I have attached a full review of the paper which is in the main well written. Statistical analysis is not used and does not need to be. I think more information re data collection is required. Please see full review.

Reviewer #2: The authors aimed to ‘identify the oral health literacy (OHL) talk in interpreter-mediated paediatric dentistry and (2) to analyze the interpreter contribution to the communication strategies’. They used mixed-methods combining VTA and CA to investigate a large number of video recorded consultations in paediatric dental practices. However, the analyses seem rather underdeveloped, and the presentation is crippled by poor quality language and structure.

The literature review has not sufficiently established the knowledge gaps. Although there is little literature on interpreter use in dentistry and paediatrics (but not non-existing), how the studies in other fields can inform the understanding of the situations in hand? The authors mention several key concepts, such as patient-centred communication, but they are insufficiently explained. It seems the authors were analysing how patient-centred (PC) approaches were used in OHL talks but it is unclear what and how established PC models were use in the analysis.

The methods section does not give sufficient detail of the CA and VTA and how VTA informed CA. When doing VTA, what model was used to identify relevant analytic moments? More detail is required to allow better understanding of how the two methods intersected in the analysis.

The results section is most under developed. It is generally descriptive and anecdotal. The presentation fails to demonstrate trend which is expected of VTA or to provide rich description of turn by turn interaction, which is expected of CA. At times the presentation was so descriptive that it lost the purpose of the overarching aim. The simple subheadings, such as PC-DI, do not provide sufficient explanation of data nor do they contribute new knowledge.

The authors have collected very rich data. The mixed methods approach has great potential to provide complex interpretation of the interactions in dental paediatrics. With more in-depth analysis this research may make valuable contribution to the literature on interpreter mediated consultations. Better presentation with improved language can help convey the message better.

6. PLOS authors have the option to publish the peer review history of their article (what does this mean?). If published, this will include your full peer review and any attached files.

Reviewer #1: No

Reviewer #2: No

---

## [Author Response · Author response to Decision Letter 0]

11 Jan 2020

Dear Prof Barbara Schouten, Academic Editor,

Thank you for considering our research article entitled “Advanced informatics understanding of clinician-patient communication: a mixed-method approach to oral health literacy talks in interpreter-mediated pediatric dentistry” (PONE-D-19-27918). Enclosed please find a marked-up copy of our manuscript that highlights changes made to the original version, and an unmarked version of our revised paper without tracked changes. With regard to the Academic Editor and Reviewers’ comments, we provide the following point by point response and explanation. 

Comments from the Academic Editor: 

Response: We thank the Editor for the advice. We have revised the manuscript to meet PLOS ONS’s style as follows. 

a. The information about the authors’ ZIP or Postal Codes, street addresses, or building/office numbers in the affiliations has been removed. 

b. The symbols about authors’ contributorship have been added. 

c. The format of the level 1 headings has been revised to be Bold type, 18pt front. 

d. The format of figure citations has been revised from “Fig. X” (Figs. Y-Z) to “Fig X” (Figs Y-Z). 

e. The format of figure titles has been revised to be Bold type. 

f. The formats of the level 2 headings and the level 3 headings have been revised to be Bold type, 16pt front and Bold type, 14pt front, respectively.

g. The format of the references with more than six authors has been revised. 

2. Please provide further details in order to justify why only 77/199 of the larger dataset were included.

Response: We thank the Editor for the concern about the selection of the 77 transcribed videos from a larger video-recorded corpus (n= 199). We have further explained the selection of the videos in the section “Material and methods - Data collection” as follows. 

“In order to ensure that the analyzed videos included all the desired elements such as “pediatric consultations”, “multi-party”, and “multilingual interpreter”, three criteria were set for the video selection from the corpus: (1) The age of patient was under 18 years old (pediatric dental consolations); (2) patient was accompanied by a parent/ caregiver (multi-party); and (3) there were Cantonese speaking children and their caregiver/ parent, and a clinical team dyad of non-local clinician (English language-in-use) with the same DSA as the interpreter (multilingual interpreter). Only the recorded videos that fulfilled all the criteria were transcribed and further analyzed in the study.”

Comments from the Reviewers: 

1. Is the manuscript technically sound, and do the data support the conclusions? The manuscript must describe a technically sound piece of scientific research with data that supports the conclusions. Experiments must have been conducted rigorously, with appropriate controls, replication, and sample sizes. The conclusions must be drawn appropriately based on the data presented. 

Response: Conversation analysis is a qualitative research method of observation to gather non-numerical data. The conclusions of the study founded on the recurrence patterns identified by the visual text analysis (VTA) and the conversation analysis of the turn-taking in different communication strategies in depth. Also, the rigorous selection of the video ensured that the study analysis aimed at the desired elements such as“pediatric consultations”, “multi-party”, and “multilingual interpreter”. Moreover, Jeffersonian transcription notation was employed to go into detail about the content of conversation and made sure that the replication of the transcript can be obtained by different researchers. Apart from the conversation presentation in text form, the plots obtained from the VTA tools helped to distinguish the different communication strategies. Hence, the conclusion was based on the presented non-numerical data (graphic recurrence pattern and video transcription).

2. Has the statistical analysis been performed appropriately and rigorously? 

Response: As mentioned, conversation analysis is a qualitative research method. Statistical analysis is not applicable for the non-numerical data in this study. 

3. Have the authors made all data underlying the findings in their manuscript fully available? The PLOS Data policy requires authors to make all data underlying the findings described in their manuscript fully available without restriction, with rare exception (please refer to the Data Availability Statement in the manuscript PDF file). The data should be provided as part of the manuscript or its supporting information, or deposited to a public repository. For example, in addition to summary statistics, the data points behind means, medians and variance measures should be available. If there are restrictions on publicly sharing data—e.g. participant privacy or use of data from a third party—those must be specified.

Response: Jeffersonian notation for the transcription of the recorded video was attached in S1 File as supporting information. The transcripts of the identified conversations in the study were also attached in S2 File as supporting information. The additional URL link of the VTA software was linked directly from the manuscript file as follows: 

“https://www.itee.uq.edu.au/research/projects/discursis”

4. Is the manuscript presented in an intelligible fashion and written in standard English? PLOS ONE does not copyedit accepted manuscripts, so the language in submitted articles must be clear, correct, and unambiguous. Any typographical or grammatical errors should be corrected at revision, so please note any specific errors here.

Response: The quality of language has been improved and grammatical errors have been corrected by SMB, CPM and OAZ who are native speakers and/or experts in Literacy. The changes are highlighted in red in the revised manuscript. 

Comments from Reviewer #1: 

1. I have attached a full review of the paper which is in the main well written. Statistical analysis is not used and does not need to be. I think more information re data collection is required. Please see full review. 

General: This is an interesting topic and it is good to see Discursis being applied to dentistry. The nature of the study, using children with their parents is also novel. I really like this paper and think it will be a valuable contribution to the literature but, as 

my comments below suggest, another revision is needed. 

Introduction: The introduction was appropriate and well referenced. It set the scene with clear research questions that were relevant to the study. There were some awkward turns of phrase that need to be addressed. 

Response: We thank the Review for the advice. We have added the subheading (Multi-party in pediatric dentistry, Multilingual interpreter, Communication strategies, Novel mixed-method approach) into the introduction to provide an organized presentation. 

2. Method: Again, the method section is clear. I think it might be useful to place the communication strategies section in the introduction as it builds on the information provided and does not relate to any methodology and data collection. 

Response: We have moved the “Communication strategies” to the Introduction. 

3. Method: I also think participant information should be in the Method section. In addition, I did not see a full procedure section beyond the little section in 2.1. I really want to see more information that outlines how the study was conducted with respect to recruitment and ages of children, education levels of parents etc. More is required here in order to better understand results.

Response: The participation information such as the language-in-use of the clinicians, patients, and parent/ primary caregivers, the age of patients, the gender of parents and parent/ primary caregivers were reported in the sub-section “Statistical description of data” in the Result section. However, the education level of parents was not recorded as it was not the focus of this study. Moreover, the sub-section “Data collection” (section 2.1) was extended to include the full study procedure to outline how the study was conducted as follows. 

“The dental team and pediatric patients/ primary caregiver (parent) dyads provided written informed consent to video recording of clinical consultations. The information of language-in-use, gender (patients and parent/ primary caregivers only), and age (patients only) was provided by dental team and/or parent/ primary caregivers via a self-complete questionnaire before video recording. In order to ensure that the analyzed videos included all the desired elements such as “pediatric consultations”, “multi-party”, and “multilingual interpreter”, three criteria were set for the video selection from the corpus: (1) The age of patient was under 18 years old (pediatric dental consolations); (2) patient was accompanied by a parent/ caregiver (multi-party); and (3) there were Cantonese speaking children and their caregiver/ parent, and a clinical team dyad of non-local clinician (English language-in-use) with the same DSA as the interpreter (multilingual interpreter). Only the recorded videos that fulfilled all the criteria were transcribed and further analyzed in the study. 

The recorded multilingual (Cantonese-English) conversations were transcribed using Transana™ software version 2.53 with Jeffersonian notation (see S1 File). All Chinese characters were parsed into English. Transcriptions and translations were verified by the team members. Data analysis of the transcribed consultations consisted of two phases.

Phase I: Visual text analysis (VTA)….

Phase II: Conversation analysis (CA)….” 

4. Results: If the results section commences at 3.2, then I think more information is required about the three principle dimensions and how they were derived. I am unclear how you decided on the actual plots. 

Response: The three principle dimensions for pattern description of Discursis™ plots were defined by the Discursis’s developers [35]. The detail of the dimensions was graphically presented in the upper right corner Fig 1. We have added more explanations as follows. 

“Time scale, direction and interaction type were the three principle dimensions used to define the multi-participant metrics of the recurrence patterns [35]. These indicated whether the recurrences were self/same speaker or other-recurrence, either following a turn (forward-direction) or recurrences related to previous turns (backward-direction) within short, medium or long period of time.

35. Angus D, Smith A, Wiles J. Human communication as coupled time series: Quantifying multi-participant recurrence. IEEE Trans Audio, Speech, Language Process. 2012;20(6):1795-807.”

5. Given we know nothing about the participants, it does not assist in understanding the generalizability of these findings.

Response: We have added the topics of the excerpts and the relevant patient information in the S2 File and the figure legends of Figs 2-7 as follows. 

“Fig 2. Inspect plot of turns 73-81 in Consultation 3 indicating patient-centered direct interpreting. “The patient’s tooth brushing behaviors” - an eight-year-old girl is here for follow-up and a possible accident causing mal-alignment of teeth.

Fig 3. Inspect plot of turns 130-145 in Consultation 3 indicating patient-centered mediated interpreting. “Suggestion for daily tooth brushing” - an eight-year-old girl is here for follow-up and a possible accident causing mal-alignment of teeth.

Fig 4. Inspect plot of turns 215-227 in Consultation 1 indicating patient-centered direct and mediated interpreting. “Dental caries prevention work after meals”- an eight-year-old boy previously treated under general anesthesia has erupting permanent teeth. 

Fig 5. Inspect plot of turns 34-66 in Consultation 11 indicating clinician-centered direct interpreting. “Inquiring about the patient’s daily tooth brushing behaviors” - a seven-year-old boy is here for his six-month review after previous treatment under general anesthesia. 

Fig 6. Inspect plots of turns 18-31 in Consultation 2 indicating clinician-centered mediated interpreting. “Giving instructions for the pulpectomy treatment” - a seven-year-old boy has missing permanent teeth.

Fig 7. Inspect plot of turns 127-146 in Consultation 6 indicating clinician-centered direct and mediated interpreting. “Giving general suggestions to the patient” – a seven-year-old-girl is brought in as she has caries.”

6. Results: It is always difficult to explain Discursis and the meaning of the diagonals in terms of engagement and time. Too much makes for a complex and unwieldy explanation. Too little leaves the reader unclear. The secret is to strike a balance. I do no think you are there yet. Some earlier explanation about how Discursis works would assist. At the moment I think you assume too much knowledge on the part of the reader. As someone who can understand these plots, I understand what you are saying but with the lack of information about participants and the lack of detail on Discursis, I think I would be in the minority. As an example, I believe you need to introduce the idea of recurrence patterns before the Results section (Line 182).

Response: We have moved the section “Discursis™ plots” (section 3.2) from Result (section 3) to Materials and methods (section 2), and added more explanations about how Discursis works as follows. 

“Discursis™ plots 

Initial, ‘long time scale plots’ as overall conceptual recurrence plots were generated to provide a global overview of both the structure and content of each consultation. Fig 1 illustrates the long-time scale recurrence pattern of a Discursis™ output for Consultation 1. Common topics such as “tooth” and “mouth” that are introduced in the initial phase are repeated and linked across the entire consultation. Time scale, direction and interaction type were the three principle dimensions used to define the multi-participant metrics of the recurrence patterns [35]. These indicated whether the recurrences were self/same speaker or other-recurrence, either following a turn (forward-direction) or recurrences related to previous turns (backward-direction) within short, medium or long period of time. Participation patterns are visualized from the color distributions and the patterns of conceptual recurrence plotting with the color, and the size of boxes in the plots corresponding to the speaker and volume per turn. The color of the lower right corner of the ‘off-diagonal’ boxes represents the speaker of the turn, while the color of the upper left corner of the off-diagonal boxes represents the speaker who shares the repeated topics or concepts. Therefore, a single or a two-colored off-diagonal box refers to the situation where a speaker repeats his or her own topics, or a speaker repeats the topics or concepts from another speaker respectively. Discursis™ inspect plots (for example, see Figs 2-7) provide the zoom-in views of the medium-time recurrence patterns from the overall plots.

…. It is recommended that the plots were read together with the relevant text transcripts in the supporting information to have a better understanding of the results.” 

7. Discussion: I returned to lines 106 to 113 and felt that these aims had not been clearly linked with the results and subsequent discussion. You talk about ‘patient participation’ but remember it is parent participation in this instance. 

Response: We have clearly defined the term “clinician-patient communication” at where it first mentioned in the manuscript as follows. 

“… clinician-patient communication (dentists and DSAs vs child patients and their parents/ primary caregivers in paediatric dental practices).”

8. Discussion: The importance of the mixed methodology was not emphasized sufficiently. Can the use of CA and Discursis be expanded?

Response: We have added more explanations about the use of CA and Discursis in the sub-section “Principal results” of Discussion as follows. 

“The participation patterns were successfully visualized by the recurrence patterns (Multi-colored or monochromatic triangular cluster) in the DiscursisTM plots. The preliminary identification of communication styles (PC and CC) can thus be achieved. The in-depth CA of the turns-at-talk among the multiparty participants (dentists and DSAs vs child patients and their parents/ primary caregivers) gave the knowledge of how they construct conversations in the different styles.”

9. Minor

a. Line 83: Hong Kong is an Asia's world city with a linguistic (should be Asian)

Response: We thank the Reviewer for the advice of the minor grammatical errors. Brand Hong Kong was launched in 2001 as a government programme designed to promote Hong Kong as "Asia’s world city" (https://www.brandhk.gov.hk/html/en/). We have added double quotes to this term.

b. Line 88: Multilingual interpreters are essential resource (needs ‘an’)

Response: We have added the word “an” in line 88 as follows. 

“Multilingual interpreters are an essential resource in between English-speaking clinicians and the Cantonese-speaking patients for the multilingualism in the dentistry in Asia [13].”

c. Line 94: …. Challenges are existed (needs changing) 

Response: We revised the sentence in line 94 as follows. 

“However, there are challenges in the medical information extraction from the clinician-patient communication.”

d. Line 110: The study also targeted to provide (needs changing) 

Response: We revised the sentence in line 110 as follows. 

“The study also aimed at providing an evidentiary account of …” 

Comments from Reviewers #2: 

1. The authors aimed to ‘identify the oral health literacy (OHL) talk in interpreter-mediated paediatric dentistry and (2) to analyze the interpreter contribution to the communication strategies’. They used mixed-methods combining VTA and CA to investigate a large number of video recorded consultations in paediatric dental practices. However, the analyses seem rather underdeveloped, and the presentation is crippled by poor quality language and structure. 

Response: We thank the Reviewers for the comments on the manuscript. Conversation analysis is a qualitative research method of observation to gather non-numerical data. Therefore, statistical analysis is not applicable for the non-numerical data in this study. The quality of language has been improved and grammatical errors have been corrected by SMB, CPM and OAZ who are native speakers and/or experts in Literacy. The changes are highlighted in red in the revised manuscript. The presentation structure has been revised as follows. 

“Introduction 

-Multi-party in pediatric dentistry

-Multilingual interpreter

-Communication strategies 

-Novel mixed-method approach 

Materials and methods

-Data collection 

-Phase I: Visual text analysis (VTA)

-Discursis™ plots 

-Phase II: Conversation analysis (CA)

Results

-Statistical description of data 

-Patient-centered (PC) communication 

-Patient-centered direct interpreting (PC-DI)

-Patient-centered mediated interpreting (PC-MI)

-Patient-centered direct interpreting and mediated interpreting (PC-DI and MI)

-Clinician-centered (CC) communication

-Clinician-centered direct interpreting (CC-DI)

-Clinician-centered mediated interpreting (CC-MI)

-Clinician-centered direct interpreting and mediated interpreting (CC-DI and MI)

-Discussion

-Comparison with prior work

-Principal results 

-Limitations

-Next steps 

-Conclusions”

2. The literature review has not sufficiently established the knowledge gaps. Although there is little literature on interpreter use in dentistry and paediatrics (but not non-existing), how the studies in other fields can inform the understanding of the situations in hand? 

Response: The key concept be reviewed in the 2nd paragraph of Introduction was “Multilingual interpreter”. Therefore, we added the subheading “Multilingual interpreter” to clarify the section. We first reviewed studies in dental [14] and medical consultations [15]. We then reviewed studies in other fields [16-20]. Finally, the need of “Multilingual interpreter” in the pediatric dentistry in Hong Kong was brought out to explain the knowledge gap [21,22]. 

14. Bridges S, Drew P, Zayts O, McGrath C, Yiu CKY, Wong HM, et al. Interpreter-mediated dentistry. Soc Sci Med. 2015;132:197-207.

15. Bolden GB. Toward understanding practices of medical interpreting: Interpreters' involvement in history taking. Discourse Stud. 2000;2(4):387-419.

16. Mason I. On mutual accessibility of contextual assumptions in dialogue interpreting. J Pragmat. 2006;38(3):359-73.

17. Baker M. Contextualization in translator-and interpreter-mediated events. J Pragmat. 2006;38(3):321-37.

18. Setton R. Context in simultaneous interpretation. J Pragmat. 2006;38(3):374-89.

19. Johnstone B. Place, globalization, and linguistic variation 2004. 65-83 p.

20. Li DC. Cantonese‐English code‐switching research in Hong Kong: A Y2K review. World Englishes. 2000;19(3):305-22.

21. Meyer B, Bührig K. Interpreting risks. Medical complications in interpreter-mediated doctor-patient communication. European J of Appl Linguist. 2014;2(2):233-53.

22. Brunner J, Chuang E, Goldzweig C, Cain CL, Sugar C, Yano EM. User-centered design to improve clinical decision support in primary care. Int J Med Inform. 2017;104:56-64.

3. The authors mention several key concepts, such as patient-centred communication, but they are insufficiently explained. It seems the authors were analysing how patient-centred (PC) approaches were used in OHL talks but it is unclear what and how established PC models were use in the analysis.

Response: We have moved the “Communication strategies” from the section “Materials and method” to “Introduction”. This subsection explains how the patient-centered communication is qualitatively defined. More explanations have been added to “Materials and method” and the subsection of “Patient-centered (PC) communication” of “Results”, which would help readers understand what and how the established PC models were used in the analysis.

4. The methods section does not give sufficient detail of the CA and VTA and how VTA informed CA. When doing VTA, what model was used to identify relevant analytic moments? More detail is required to allow better understanding of how the two methods intersected in the analysis.

Response: We have moved the section “Discursis™ plots” (section 3.2) from Result (section 3) to Materials and methods (section 2), and added more explanations about how VTA (Discursis) works and how it intersected with CA as follows.

“Discursis™ plots 

Initial, ‘long time scale plots’ as overall conceptual recurrence plots were generated to provide a global overview of both the structure and content of each consultation. Fig 1 illustrates the long-time scale recurrence pattern of a Discursis™ output for Consultation 1. Common topics such as “tooth” and “mouth” that are introduced in the initial phase are repeated and linked across the entire consultation. Time scale, direction and interaction type were the three principle dimensions used to define the multi-participant metrics of the recurrence patterns [35]. These indicated whether the recurrences were self/same speaker or other-recurrence, either following a turn (forward-direction) or recurrences related to previous turns (backward-direction) within short, medium or long period of time. Participation patterns are visualized from the color distributions and the patterns of conceptual recurrence plotting with the color, and the size of boxes in the plots corresponding to the speaker and volume per turn. The color of the lower right corner of the ‘off-diagonal’ boxes represents the speaker of the turn, while the color of the upper left corner of the off-diagonal boxes represents the speaker who shares the repeated topics or concepts. Therefore, a single or a two-colored off-diagonal box refers to the situation where a speaker repeats his or her own topics, or a speaker repeats the topics or concepts from another speaker respectively. Discursis™ inspect plots (for example, see Figs 2-7) provide the zoom-in views of the medium-time recurrence patterns from the overall plots. The qualitative analysis of the turns-at-talk among the multiparty participants was conducted in Phase II using CA.”

5. The results section is most under developed. It is generally descriptive and anecdotal. The presentation fails to demonstrate trend which is expected of VTA or to provide rich description of turn by turn interaction, which is expected of CA. At times the presentation was so descriptive that it lost the purpose of the overarching aim.

Response: Descriptive and anecdotal are characteristics for results of Qualitative research (Mayes et al 2007). We have added more explanations to emphasize the importance of CA and Discursis in this study in the sub-section “Principal results” of Discussion as follows. 

“The participation patterns were successfully visualized by the recurrence patterns (Multi-colored or monochromatic triangular cluster) in the DiscursisTM plots. The preliminary identification of communication styles (PC and CC) can thus be achieved. The in-depth CA of the turns-at-talk among the multiparty participants (dentists and DSAs vs child patients and their parents/ primary caregivers) gave the knowledge of how they construct conversations in different styles.”

Mayes LE, Fonagy PE, Target ME. Developmental science and psychoanalysis: Integration and innovation: Karnac Books; 2007.

6. The simple subheadings, such as PC-DI, do not provide sufficient explanation of data nor do they contribute new knowledge.

Response: We have revised the subheadings as follows. 

“-Patient-centered (PC) communication 

-Patient-centered direct interpreting (PC-DI)

-Patient-centered mediated interpreting (PC-MI)

-Patient-centered direct interpreting and mediated interpreting (PC-DI and MI)

-Clinician-centered (CC) communication

-Clinician-centered direct interpreting (CC-DI)

-Clinician-centered mediated interpreting (CC-MI)

-Clinician-centered direct interpreting and mediated interpreting (CC-DI and MI)”

7. The authors have collected very rich data. The mixed methods approach has great potential to provide complex interpretation of the interactions in dental paediatrics. With more in-depth analysis this research may make valuable contribution to the literature on interpreter mediated consultations. Better presentation with improved language can help convey the message better.

Response: Your comments are very much appreciated. Not many readers are familiar with Discursis, so much space of this paper is devoted to help the readers understand the results. Concerning the focus and length of this paper, more analyses will be conducted in the future.

We hope you are satisfied with our responses. 

Best regards,

Dr. Hai Ming Wong 

Clinical Associate Professor

Department of Paediatric Dentistry, Faculty of Dentistry, The University of Hong Kong

Address: PDO, 2/F Prince Philip Dental Hospital, 34 Hospital Road, HK

Fax: 25593803

Tel: 28590261

Email: wonghmg@hku.hk

---

## [Decision Letter · Decision Letter 1]

27 Feb 2020

PONE-D-19-27918R1

Advanced informatics understanding of clinician-patient communication: a mixed-method approach to oral health literacy talks in interpreter-mediated pediatric dentistry

PLOS ONE

Dear dr. Hai Ming Wong,

Thank you for submitting your manuscript to PLOS ONE. After careful consideration, we feel that it has merit but does not fully meet PLOS ONE’s publication criteria as it currently stands. Therefore, we invite you to submit a revised version of the manuscript that addresses the points raised during the review process.

Both reviewers were in general satisfied with your revision. However, they have some minor comments that should be addressed; in particular, as suggested by reviewer 1, the discussion-section can be enhanced a bit. In addition, I would add the fact that you did not assess parent's educational level as a limitation of your study, because there is a clear link established in previous research between educational level on the one hand, and health literacy and patient participation on the other hand. Please briefly address this as well in your discussion section.  

We would appreciate receiving your revised manuscript by March 29th 2020. To enhance the reproducibility of your results, we recommend that if applicable you deposit your laboratory protocols in protocols.io, where a protocol can be assigned its own identifier (DOI) such that it can be cited independently in the future. For instructions see: http://journals.plos.org/plosone/s/submission-guidelines#loc-laboratory-protocols

We look forward to receiving your revised manuscript.

Kind regards,

Barbara Schouten

Academic Editor

PLOS ONE

Reviewers' comments:

Reviewer's Responses to Questions

**Comments to the Author**

1. If the authors have adequately addressed your comments raised in a previous round of review and you feel that this manuscript is now acceptable for publication, you may indicate that here to bypass the “Comments to the Author” section, enter your conflict of interest statement in the “Confidential to Editor” section, and submit your "Accept" recommendation.

Reviewer #1: All comments have been addressed

Reviewer #2: (No Response)

2. Is the manuscript technically sound, and do the data support the conclusions?

Reviewer #1: Yes

Reviewer #2: Partly

3. Has the statistical analysis been performed appropriately and rigorously? 

Reviewer #1: N/A

Reviewer #2: Yes

4. Have the authors made all data underlying the findings in their manuscript fully available?

Reviewer #1: Yes

Reviewer #2: Yes

5. Is the manuscript presented in an intelligible fashion and written in standard English?

Reviewer #1: Yes

Reviewer #2: Yes

6. Review Comments to the Author

Reviewer #1: I congratulate the authors on the revisions. I think the paper is much clearer than before and is a valuable contribution to the literature.

I think there may be a typo on line 262 where the authors say, “The greed-blue and pink-blue……”. Should be this green-blue?

I have one further request which may reflect my discipline, but I would like to see the aims explicitly revisited in the Discussion. In particular the second aim concerning PC-DI, PC-MI, CC-DI and CC-MI.

They are clearly described in the results but then not further discussed later. I think you could do more in the discussion to clarify and round off the paper.

Reviewer #2: The revision has improved significantly. The language flows much smoother and is easy to read. The details about the two-phased analysis provide clarity on how the mixed methods approach was implemented in this study, which was one of my questions.

I feel the authors do not need to refer to patient-centred communication in this study. I find it problematic to determine whether the interaction is patient-centred purely based on the level of engagement of the parent and the patient in the interaction. It is well established that patients speak less in information giving/explaining stage of a consultation, and this does not mean the clinician has taken a non-patient-centred approach. This is true in the authors' study. I read the transcript, where I can see a lot of good practices of the clinicians that would be deemed as patient centred. For example, they adapted the language when giving instructions to make it simple and digestible for children.

The focus of this study is to unveil the patterns. This is already achieved without referencing patient-centred communication.

I think the transcript can be improved. The authors claim that they used Jeffersonian conventions; however, the presentation of the transcripts are not strictly following the conventions. For example, pauses/gaps are usually marked like e.g. (0.96), with the number inside to indicate the length of a pause. Overlapping speeches should be aligned with each other marked with [.

There are still some typos in the manuscript. Here are some that I have noticed:

1. I don’t understand the sentence in lines 90-92. Maybe rephrase.

2. 98—healthcare communication not health sciences communication

3. 137—what’s the average length of consultation? What’s the total hours of recording?

4. 189—may be delete ‘with turns-at-talk overlap’.

5. 193—Spell out MI.

7. PLOS authors have the option to publish the peer review history of their article (what does this mean?). If published, this will include your full peer review and any attached files.

Reviewer #1: No

Reviewer #2: No

---

## [Author Response · Author response to Decision Letter 1]

1 Mar 2020

Dear Prof Barbara Schouten, Academic Editor,

Thank you for considering our research article entitled “Advanced informatics understanding of clinician-patient communication: a mixed-method approach to oral health literacy talks in interpreter-mediated pediatric dentistry” (PONE-D-19-27918R1). Enclosed please find a marked-up copy of our manuscript that highlights changes, and an unmarked version of our revised paper without tracked changes. With regard to the Academic Editor and Reviewers’ comments, we provide the following point by point response and explanation. 

Comments from the Academic Editor: 

1. Both reviewers were in general satisfied with your revision. However, they have some minor comments that should be addressed; in particular, as suggested by reviewer 1, the discussion-section can be enhanced a bit.

Response: We thank the reviewers’ agreements on the previous revisions. We have revised the manuscript to meet the reviewers’ new comments. 

2. In addition, I would add the fact that you did not assess parent's educational level as a limitation of your study, because there is a clear link established in previous research between educational level on the one hand, and health literacy and patient participation on the other hand. Please briefly address this as well in your discussion section.

Response: We thank the Editor for the concern about the effect of educational level of the patients in the study. We have added the discussion on the study limitations and the study contribution to future work in the sub-sections “Discussion – Limitations” and “Discussion – Next steps” as follows. 

“The main focuses of the study were to investigate the recurrence patterns of OHL talks in interpreter-mediated pediatric dentistry and to qualitatively explore the content of the talks. “Language-in-use” of the multi-parties was the major criterion in the selection of the OHL talks including a multilingual interpreter. Despite the randomness of patient background, the same DAS (multilingual interpreter) was involved in all the selected videos to reduce the effects of clinicians’ background. However, the effects of parents and patients’ socio-economic status [54] such as educational level on communication strategies, or the effects of communication strategies on the outcomes [3, 4] such as their satisfaction, were not investigated and discussed in this study. Those effects need to be further quantitatively investigated for designing a guideline on good health communication practices.”

54. Willems S, De Maesschalck S, Deveugele M, Derese A, De Maeseneer J. Socio-conomic status of the patient and doctor–patient communication: does it make a difference? Patient Educ Couns. 2005;56(2):139-46. 

“In an ideal situation, a better clinician-patient communication can be achieved by all multi-parties with higher educational levels. Although the educational levels of patients or their parents are not controllable, the communication skills of dental professionals are educable. The popularization of the use of VTA……” 

Comments from Reviewer #1: 

1. I congratulate the authors on the revisions. I think the paper is much clearer than before and is a valuable contribution to the literature. 

Response: We thank the Review for the agreement on the revision and the contribution of the paper. 

2. I think there may be a typo on line 262 where the authors say, “The greed-blue and pink-blue……”. Should be this green-blue?

Response: We have revised the sentence in line 262 as follows. 

“The green-blue and pink-blue ……” 

3. I have one further request which may reflect my discipline, but I would like to see the aims explicitly revisited in the Discussion. In particular the second aim concerning PC-DI, PC-MI, CC-DI and CC-MI. They are clearly described in the results but then not further discussed later. I think you could do more in the discussion to clarify and round off the paper.

Response: We have explicitly revisited the two research questions (aims) of the study in the sub-sections “Discussion – Principal results” to clarity and round off the paper as follows. 

“With regard to the 1st research question ‘where’ were the OHL talks, the clinician-dominated and multi-party participation patterns were successfully visualized by the recurrence patterns of monochromatic and multi-colored triangular cluster in the Discursis™ plots, respectively. The former pattern was preliminarily identified to be CC communication style, while the latter was PC communication style. Conventionally, it was hard to quantitatively assess the level of interactive engagement of participants in conversations. This increased the difficulties in identifying different communication styles. The study results showed that the application of the VTA can be an alternative to the preliminary identification of the communication styles, and a valuable approach of the PC communication studies. Subsequently, the in-depth CA of the turns-at-talk among the multiparty participants (dentists and DSAs vs child patients and their parents/ primary caregivers) gave the knowledge of ‘how’ multi-party, multilingual pediatric dental consultations were interactively managed – the 2nd research question. The two communication styles (PC and CC) can then be further divided in to three strategies (-DI, -MI, and -DI and MI). DI illustrated how the DSA (multilingual interpreter) cooperated with dentists, and MI illustrated how the interpreter tried to build a connection with the patients/ primary caregivers. The mix of DI and MI further showed how the multi-party, multilingual pediatric dental consultations were constructed by the interactions of the participants. The importance of multilingual interpreter was demonstrated in both unveiled communication strategies.” 

Comments from Reviewers #2: 

1. The revision has improved significantly. The language flows much smoother and is easy to read. The details about the two-phased analysis provide clarity on how the mixed methods approach was implemented in this study, which was one of my questions. 

Response: We thank the Review for the agreement on the revisions and the study approach. 

2. I feel the authors do not need to refer to patient-centred communication in this study. I find it problematic to determine whether the interaction is patient-centred purely based on the level of engagement of the parent and the patient in the interaction. It is well established that patients speak less in information giving/explaining stage of a consultation, and this does not mean the clinician has taken a non-patient-centred approach. This is true in the authors' study. I read the transcript, where I can see a lot of good practices of the clinicians that would be deemed as patient centred. For example, they adapted the language when giving instructions to make it simple and digestible for children. The focus of this study is to unveil the patterns. This is already achieved without referencing patient-centred communication.

Response: Since the introduction of the PC communication is important to differentiate between the two unveiled recurrence patterns (monochromatic and multi-colored triangular cluster) in the Discursis™ plots of the study, we need to keep the refereeing process. In addition, one of the difficulties in identifying communication styles is that communication styles cannot be easily quantitatively defined. It is the reason why visual text analysis (VTC) was needed in this study and how this study contributed to studies of PC communication. We have added more explanations about importance of the study results to the field of PC communication in the sub-sections “Discussion – Principal results” as follows. 

“With regard to the 1st research question ‘where’ were the OHL talks, the clinician-dominated and multi-party participation patterns were successfully visualized by the recurrence patterns of monochromatic and multi-colored triangular cluster in the Discursis™ plots, respectively. The former pattern was preliminarily identified to be CC communication style, while the latter was PC communication style. Conventionally, it was hard to quantitatively assess the level of interactive engagement of participants in conversations. This increased the difficulties in identifying different communication styles. The study results showed that the application of the VTA can be an alternative to the preliminary identification of the communication styles, and a valuable approach of the PC communication studies.”

3. I think the transcript can be improved. The authors claim that they used Jeffersonian conventions; however, the presentation of the transcripts are not strictly following the conventions. For example, pauses/gaps are usually marked like e.g. (0.96), with the number inside to indicate the length of a pause. 

Response: In Jeffersonian intonation marks, the time of the pauses is accurate to 0.5s ((.): noticeable pause shorter than 0.5 s; (~n): timed pause in n integer second). We have synchronized the intonations (e.g. (short pause), (pause), (SP), and (0.1)) to the required intonation (.) in the revised S2 File. The revise S2 file is also attached for the reference. 

4. Overlapping speeches should be aligned with each other marked with [.

Response: The focus of the study was the text content of the conversation. Also, the bilingual transcripts will become messy and plenty of empty spaces will be generated, if the content of the conversation is forced to be aligned. We therefore decided not to align the text of the overlapping speeches in the transcripts. 

5. There are still some typos in the manuscript. Here are some that I have noticed:

a. I don’t understand the sentence in lines 90-92. Maybe rephrase.

Response: We have rephrased the sentence in lines 90-92 as follows. 

“Research in ad hoc interpreting has raised concerns regarding the communication of the risks in medical complications [21] and the improvements in clinical shared decision-making [22].”

b. 98—healthcare communication not health sciences communication

Response: We have revised the word “healthcare” in line 98 as follows. 

“……classifications in healthcare communication.”

c. 137—what’s the average length of consultation? What’s the total hours of recording?

Response: The average length and the total hours of the larger video-recorded corpus (n=199), including the videos not fulfilling the selection criteria, were irrelevant and were excluded from the study analysis. The average length (12.62 mins) and total hours (12.62*77 = 971.74 mins or 16.2 hours) of the selected videos have been already presented in lines 198-199 as follows. 

“The average length and turns-at-talk of the 77 transcribed video recordings were 12.62 minutes and 290 turns, respectively.”

d. 189—may be delete ‘with turns-at-talk overlap’.

Response: We have deleted the words “with turns-at-talk overlap” in line 189. 

e. 193—Spell out MI.

Response: The short form of the term “mediated interpreting (MI)” was first mentioned in line 84 of the paper. We therefore keep the use of the short form MI in line 193. 

We hope you are satisfied with our responses. 

Best regards,

Dr. Hai Ming Wong 

Clinical Associate Professor

Department of Paediatric Dentistry, Faculty of Dentistry, The University of Hong Kong

Address: PDO, 2/F Prince Philip Dental Hospital, 34 Hospital Road, HK

Fax: 25593803

Tel: 28590261

Email: wonghmg@hku.hk

---

## [Editor Report · Decision Letter 2]

4 Mar 2020

Advanced informatics understanding of clinician-patient communication: a mixed-method approach to oral health literacy talks in interpreter-mediated pediatric dentistry

PONE-D-19-27918R2

Dear Dr. Hai Ming Wong,

We are pleased to inform you that your manuscript has been judged scientifically suitable for publication and will be formally accepted for publication once it complies with all outstanding technical requirements.

With kind regards,

Barbara Schouten

Academic Editor

PLOS ONE

---

## [Editor Report · Acceptance letter]

6 Mar 2020

PONE-D-19-27918R2 

Advanced informatics understanding of clinician-patient communication: a mixed-method approach to oral health literacy talk in interpreter-mediated pediatric dentistry 

Dear Dr. Wong:

I am pleased to inform you that your manuscript has been deemed suitable for publication in PLOS ONE. Congratulations! Your manuscript is now with our production department. 

With kind regards,

on behalf of

Dr. Barbara Schouten 

Academic Editor

PLOS ONE